# Appendage Regeneration in Vertebrates: What Makes This Possible?

**DOI:** 10.3390/cells10020242

**Published:** 2021-01-27

**Authors:** Valentina Daponte, Przemko Tylzanowski, Antonella Forlino

**Affiliations:** 1Biochemistry Unit, Department of Molecular Medicine, University of Pavia, via Taramelli 3/B, 27100 Pavia, Italy; valentina.daponte01@universitadipavia.it; 2Skeletal Biology and Engineering Research Center, Department of Development and Regeneration, University of Leuven, 3000 Leuven, Belgium; przemko@kuleuven.be; 3Department of Biochemistry and Molecular Biology, Medical University of Lublin, 20-059 Lublin, Poland

**Keywords:** appendage regeneration, WNT/β catenin, FGF, lizard, salamander, zebrafish, stem cells, differentiation, dedifferentiation, signaling pathways

## Abstract

The ability to regenerate amputated or injured tissues and organs is a fascinating property shared by several invertebrates and, interestingly, some vertebrates. The mechanism of evolutionary loss of regeneration in mammals is not understood, yet from the biomedical and clinical point of view, it would be very beneficial to be able, at least partially, to restore that capability. The current availability of new experimental tools, facilitating the comparative study of models with high regenerative ability, provides a powerful instrument to unveil what is needed for a successful regeneration. The present review provides an updated overview of multiple aspects of appendage regeneration in three vertebrates: lizard, salamander, and zebrafish. The deep investigation of this process points to common mechanisms, including the relevance of Wnt/β-catenin and FGF signaling for the restoration of a functional appendage. We discuss the formation and cellular origin of the blastema and the identification of epigenetic and cellular changes and molecular pathways shared by vertebrates capable of regeneration. Understanding the similarities, being aware of the differences of the processes, during lizard, salamander, and zebrafish regeneration can provide a useful guide for supporting effective regenerative strategies in mammals.

## 1. Introduction

Regeneration is the capacity of an organism to regrow a part of the body after injury. Many species of anamniote vertebrates are capable of spectacular accomplishment of regeneration. Among them, the best known is salamander, which is able to restore limb, tail, eye, jaw, and heart [1]. Regeneration is also present in various members of the class of actinopterygians [2]. The adult zebrafish (*Danio rerio*), a small teleost, exhibits remarkable regeneration of fins, central nervous system structures, or entire organs including heart, pancreas, liver, and kidney [3]. The ability to activate the regeneration morphogenetic program is almost completely lost mainly in terrestrial animals, such as amniotes, in which nearly all structures are unable to regenerate, with some exceptions such as the regeneration of digit tips in children [4] and young mice [5], ear pinna in the spiny mouse [6], or the antler regeneration in deer [7]. Lizards are the only terrestrial amniotes that can regenerate a structure long and complex as the tail [8].

The distribution of the regenerative capacity along the phylogenetic tree is haphazard and a common pattern is difficult to identify (Figure 1). Regeneration appeared very early during evolution, likely coinciding with the origin of multicellularity [9]. The similarities between regeneration and development may suggest that the former originated as an epiphenomenon of an ontogenetic program that an organism accesses and employs when a structure is lost. Based on these observations, regeneration could be an ancestral property, lost due to higher energetic maintenance costs (for a detailed review, see [10]).

In salamanders, several genes are associated with the regeneration ability, one of them is the one coding for the glycerophosphatidyl (GPI)-anchored cell surface Prod1, which mediates positional memory through a proximal-distal gradient of cell adhesion [11]. Despite a protein with similar functions being recently identified in the regenerating tail of the lizard *Gekko japonicus* [12], the Prod1 gene has not been found in any other species [13].

The ability to regenerate paired and unpaired fins was documented in many ray-finned fish (Actinopterygii). Teleosts possess a wide range of regenerative abilities, from complete absence of regeneration in some members of the family Blennidae [14] to a high regenerative ability of zebrafish. However, this trait is limited to the bony fin rays and does not include the proximal structures such as musculature and endoskeleton. The only exception are Polypteriformes, one of the basal lineages of Actinopterygii [15,16]. The strong regenerative potential found in this taxon suggests that regeneration in this case has emerged early during evolution, long before the appearance of Teleostei. In terrestrial amniotes, the evolution of a strong adaptive immune system could have favored a more defensive role of macrophages during the early phases of regeneration to the expenses of their ability of tissue remodeling [17]. The extreme “non-lethal predation” of lizards’ tails in the wild could instead explain the presence of a regenerative program in these amniotes [18].

Many factors have contributed to the loss of regeneration in the evolution, and further phylogenetic and comparative studies are necessary to unravel the mechanisms regulating the process. The investigation of the regeneration of different organs, from the simple tail to the more complex limb in different organisms, represents a powerful tool to shed light on the process. This review will examine appendage regeneration in vertebrates, with a specific focus on tail regeneration in lizards, limb regeneration in salamanders, and fin regeneration in zebrafish. We will describe common cellular mechanisms across the regenerative process in these animals and investigate differences that could have led to the loss of regenerative capacity in mammals.

## 2. Experimental Approaches

The study of the regenerative process in lizard, salamander, and zebrafish, expanded in the last years by the development of new experimental tools, provided a better understanding of the regeneration mechanisms in these species pointing out similarities and differences.

The ability of lizards to spontaneously lose their tail and then regrow it as an anti-predation strategy (known as “autotomy”) made lizards one of the animal models used to study healing and regeneration. Autotomy is often exploited in laboratories to induce less invasive traumas and avoid surgery, so that simply pinching the tail causes the tail to drop [8]. After tail loss, tissues are easily accessible, allowing manipulation and study in vivo. In recent years, several lizard genomes have been characterized, including that of the green lizard *Anolis carolinensis*, the primary candidate for regeneration studies [19]. This allowed to carry out genomic and transcriptomic analysis of genes involved in wound healing, cell proliferation and inflammatory-immune response to injury [20,21,22,23,24]. Nonetheless, transgenic or knockout/knockdown lizard models are still difficult to generate due to the complexity of accessing zygotes for genetic manipulation. With the advent of CRISPR-Cas9-mediated gene editing, this will likely change and recently, the first gene-edited *Anolis*, deficient for tyrosinase gene, has been created [25]. Given their close relation to mammals, as the only amniote model system for regeneration, the establishment of transgenic and mutant lizard models may help to clarify the loss of regenerative capacities in mammals during evolution.

Salamanders are another example of vertebrates with remarkable regeneration capabilities, being able not only to regenerate tail, but also limbs, neural cells [26], and heart [27]. Various salamander species captured the interest of regenerative research. Among them, the paedomorphic axolotl (*Ambystoma mexicanum*) has been investigated since the 1860s [28], emerging as the first salamander model for regenerative studies. Recently, a newt model (*Pleurodeles waltl*), sharing the same regeneration abilities and the short life cycle of the axolotl, has been established [29]. Given the feasibility for functional genetic approaches provided by both models, a large number of tools, including transgenesis and gene silencing, has begun to accumulate [30,31,32]. Axolotl and *Pleurodeles* genomes are large and rich in long terminal repeat (LTR) retrotransposons [33]. Their genomes are now sequenced and annotated [29,34,35], paving the ways for deeper genetic approaches, such as mutagenesis [36], transcriptomics [37], and genetic lineage tracking [38].

Zebrafish (*Danio rerio*), a freshwater teleost, has become one of the most widely used vertebrates for the study of regeneration. It has several experimental advantages, such as ease of manipulation and short developmental cycle. Its genome was sequenced and revealed about 70% coding sequence conservation to humans [39]. Importantly, zebrafish can regenerate several organs such as fin, heart, kidney, liver or brain [3]. The rapid regeneration of zebrafish caudal fin has made them one of the most powerful tools for regenerative studies, especially since the transcriptomics approaches are possible [40,41,42], and transgenesis and morpholinos techniques are readily available [43,44,45,46]. The recently established CRISPR-Cas9 gene editing additionally increased the value of this model. Zebrafish are also particularly suitable for small-molecule drug application, which, in skeletal field, takes enormous advantage of tail regeneration [47].

To summarize, lizard, salamander, and zebrafish offer a variety of experimental approaches for the investigation of the regeneration processes (Table 1), which will represent a huge opportunity to expand the understanding of appendage regeneration in vertebrates.

## 3. Common Morphological Aspects of Early Regenerative Response

Lizard tail, salamander limb, and zebrafish fin regeneration are examples of epimorphosis. The term epimorphic regeneration, coined by Thomas H. Morgan in 1901 [57], describes the restoration of a part of the organism without remodeling, characterized by the formation of the blastema. This mass of undifferentiated cells at the wound site mediates tissue differentiation [58,59]. Epimorphosis is characterized by a high degree of cellular differentiation and stands in contrast with morphallaxis, usually observed in hydras, in which blastema is not formed and the majority of regeneration is taking place by reorganization of the remaining parts of the body [58,59].

Lizard, salamander, and zebrafish share several common morphological aspects of epimorphic regeneration that distinguish this process from the limited regeneration in mammals. Some differences were however identified. For example, lizards, unlike salamanders, are unable to regenerate amputated limbs. Also, after autotomy or amputation, lizard tail undergoes an “imperfect regeneration” in which the bony vertebral column is replaced with a hollow cone of cartilage and the fracture planes, that are structures evolved to allow autotomy at precise locations, are not regenerated [60]. On the contrary, salamanders and zebrafish can restore the original appendage. Lastly, while lizard and salamander appendage regeneration occur following cartilage formation, zebrafish caudal fin rays are directly generated as bone.

The regenerative process begins immediately after an appendage loss and consists of three phases: wound healing, blastema formation and regenerative outgrowth (Figure 2).

Initially, after autotomy or amputation, a blood clot forms. Lizard’s tail stump has evolved the presence of vascular sphincters located proximal to fracture planes that contract to avoid excessive bleeding and to facilitate hemostasis [61]. Within a few hours, epithelial cells begin to migrate to cover the wound, forming an epithelial layer called the wound epidermis. The integrity of the wound epidermis is essential to accomplish regeneration in lizard and salamander. Impeding its formation by excision or suture prevents the appendage to regenerate [62,63]. Repetitive amputations seem to trigger a persistent wound healing response, and result in a decline in regenerative fidelity in lizard and salamander [64,65]. In the newt *Notophthalmus viridescens*, repeated amputation of the limb can lead to severe morphological abnormalities [66]. To the contrary, in zebrafish, repeated amputations do not perturb regeneration and instead lead to an increase in the dermal bone thickness [67].

After several days, in salamander and zebrafish, the thick wound epidermis forms a specialized structure, the apical epidermal cap (AEC) that acts as a layer of signaling cells coordinating the regenerative process [61,68]. For this reason, AEC needs to be continuously in contact, and to communicate with, the underneath mesenchymal cells. Interposition of dermis or the formation of basement membrane between the two structures inhibits blastema formation and regeneration [69]. This property suggests that diffusible signaling factors are produced in the AEC and signal to the underlying mesenchyme. In lizards, this thick epithelium has a different morphology, owing a discontinuous basement membrane, and is often referred to as an apical epidermal peg (AEP) [70].

As quickly as the wound epidermis thickens, the tissues below undergo degeneration and histolysis. The dynamic remodeling of extracellular matrix (ECM) plays a crucial role in this process. The secretion of acid hydrolases and proteinases of the matrix metalloproteinases (MMPs) family at the level of the basal layer of the wound epidermis helps degrading ECM components, such as collagens and laminins. This process is likely facilitating cell migration and cell communication by disturbing basement membrane re-assembly [71,72,73]. MMPs also favor the elimination of cellular residues and debris generated by tissue destruction and by the bactericidal activity of neutrophils and macrophages [69]. In lizard and salamander, the degradation involves primarily bone and muscle [74,75]. In zebrafish, in which the caudal fin is mainly composed of bone, the establishment of a cryoinjury-mediated tissue damage triggers a similar osteolytic process [76]. The degeneration of stump tissues, or “histolysis”, favors the release of progenitor cells that will contribute to the formation of blastema. Comparative anatomy of regenerating appendages in lizard, salamander, and zebrafish is depicted in Figure 3.

## 4. Role and Origin of Blastema: Dedifferentiation versus Stem Cells

Blastema is the key component of the epimorphic process and the primary source of differentiated cells responsible for appendage regeneration. Blastema formation occurs within 10–15 days post-amputation in most lizard species [20,77]. After tail autotomy of the leopard gecko *Eublepharis macularius*, blastema can be identified after 3 days, and continues to expand until 8 days post-autotomy [78]. In adult newts, blastema cells appear within 4-5 days post-amputation [69] and accumulate around 7 days post-amputation in axolotl [79,80]. In zebrafish, blastema can be identified at day 2 post-amputation (Figure 2) [68,81].

Despite the role of blastema as the necessary and sufficient structure for the progression of regeneration, its origin is still controversial and debated. Two competing hypotheses have emerged to define the source of activated progenitor cells within the blastema. The first one points to resident stem cells characterized by self-renewal potency and the ability to produce one or more differentiated cell types [82]. The second hypothesis foresees a cell conversion event by which differentiated cells dramatically change their identity. One of the mechanisms associated with this conversion is dedifferentiation, in which differentiated cells lose their specialized function and revert to a less-differentiated stage with higher potential, akin to the events taking during the generation of induced pluripotent stem cells from dermis [83]. The other possible mechanism is direct transdifferentiation, in which differentiated cells switch lineage and are converted into another cell type within or across germ layers. Transdifferentiation is naturally observed during lens regeneration in newts where removal of the lens induces the formation of a new lens derived from the cells of the iris [84,85]. Direct transdifferentiation occurs without going through a stem/progenitor-like cell [86,87,88].

In lizard tail regeneration fails when the amputation occurs at its base, in the non-autotomous region. This phenomenon has been hypothesized as caused by the depletion of stem cells after the basal amputation of the tail [77]. Indeed, histological, immunocytochemical, and autoradiographic studies indicate the presence of slowly cycling, putative resident stem/progenitor cells in various tissues of lizards’ original tail, including satellite cells of muscles and ganglia, and chondroblasts localized in intervertebral cartilages and vertebrae [89,90]. Following injury, these cells can activate, as demonstrated by the immune detection of telomerase and c-myc, and migrate to contribute to the blastema [91,92]. Transcriptomic and histological works on *Anolis carolinensis* tail regeneration led to the identification of a population of PAX7+ satellite cells, pointing to a stem-cell mediated process [20,93].

On the other hand, a study on the lizard *Podarcis muralis* suggests that the activation of a dedifferentiation process could be linked to a degree of tissue damage. After 3-4 days post-autotomy, fragmented myofibers showed an infiltration by macrophages indicating phagocytosis and gave origin to small cells with euchromatic nuclei, interpreted as dedifferentiated muscle fibers. Similarly, the spinal cord, fibrocytes of the dermis or inter-muscle connective tissues, adipose and cartilaginous tissues, following damage, gave rise to viable, proliferating cells [94]. A recent lineage tracking study on the mourning gecko (*Lepidodactylus lugubris*) indicates that during tail regeneration, differentiated cartilage cells contribute to muscle and differentiated muscle cells contribute to cartilage in the regenerated tissue, suggesting that dedifferentiation or even transdifferentiation could play a role [95]. The contribution of muscle to cartilage cells was also described during axolotl tail regeneration [96]. These findings are in contrast with what was reported for limb regeneration. Using a GFP transgene to track limb tissues, Kragl et al. showed that limb blastema cells are lineage-restricted and only capable to give origin to cells from the same germ layer. Dermis was able to form cartilage and tendons, derived from the lateral plate mesoderm, but not muscle, that derives from the presomitic mesoderm. In the same way, muscle was able to give origin to only muscle and not cartilage [30]. These contradictory findings could, in part, be explained by the embryonic differences between tail and limb buds, with the former having a pluripotent nature and being able to give origin to all the three germ layers [97]. Differences are also seen when comparing blastemas of *Anolis* lizards and leopard gecko *Eublepharis macularius.* In the Anolis lizard, no cell proliferation was detectable leading some authors to question whether this was a real blastema [20,78,93], whereas the leopard gecko blastema was reported to be rich in actively proliferating cells [72,98,99]. A recent study by Lozito and Tuan suggests that Anolis tail regeneration could require cell populations of different origin. This study was aimed to clarify why proximal regenerated lizard tail skeleton undergoes hypertrophy and endochondral ossification, whereas the distal regenerate remains cartilaginous. The study shows that the proximal regenerated lizard tail skeleton is derived from progenitor/stem cells within the vertebrae periosteum in response to BMP and Ihh signaling, whereas more distal regions are derived from multipotent blastema cells and respond to Shh signals [100].

Different contributions to blastema are also found within closely related species. In the newt *Notophthalmus viridescens*, myofiber dedifferentiation is required for blastema formation and generates PAX7− cells that will later give rise to new muscle in the regenerated tissue. In axolotl, after amputation, myofibers give rise to neither proliferative cells in the blastema nor to regenerated muscle. Instead, the primary contributors to blastema are PAX7+ satellite cells [101,102]. However, a combination of the two mechanisms cannot be excluded. A recent single-cell RNA-sequencing study on axolotl limb identified the presence of PAX7+ cells and, likely, tendons progenitor cells together with a population of multipotent fibroblasts able to form joints, cartilage, and bone of the regenerate [103]. Whether fibroblasts derive from rare specialized progenitor populations or from general activation of already differentiated fibroblasts, remains to be determined.

Blastema formation in zebrafish caudal fin occurs mainly, if not exclusively, by dedifferentiation. Studies employing different cell-tracking methods revealed that cells followed lineage restriction. Knopf et al. demonstrated that osteoblasts temporarily dedifferentiate in response to amputation. In the caudal fin, a strong expression of the mature osteoblast marker *osteocalcin* was detected in the bony rays. Following the amputation and ensuing regeneration, sequential induction of *runx2*, pre-osteoblasts marker at 2 days post-amputation, *osterix*/*sp7*, marker for intermediate differentiation of osteoblasts at 3 days post-amputation and *osteocalcin* at 6–7 days post-amputation were detected [44]. The same authors, using *Cre*-loxP-based fate-mapping, demonstrated that dedifferentiating osteoblasts enter fin blastema and give rise only to osteoblasts during zebrafish fin regeneration. They implemented a double transgenic fish line carrying tamoxifen-inducible CreERT2 under the *osterix* promoter (osterix:CreERT2) in conjunction with a floxed DsRed-Stop cassette under the control of a heat shock promoter expressing GFP after the excision of the DsRed-Stop cassette (hsp70:loxP DsRed2 loxP nlsEGFP). During regeneration, GFP+ cells were found within the blastema, forming new osteoblasts, without significantly contributing to other lineages [44]. A similar approach was used by another group and led to the same results [104].

Tu and Johnson generated mosaic fish by inserting a XenEF1α:GFP transgene by Tol2 transposition into 1–2 cell embryos [105]. The EF1α promoter, driving GFP expression, was silenced in adult tissues, and its expression was restricted to the blastema during fin regeneration, providing a useful tool to visualize differentiating cells in regenerative tissues [106]. Co-occurrence analysis for GFP labeled clones showed that caudal fin is formed by 9 distinct lineage classes: the unipotent osteoblast, fibroblast, glial, iridophore, epidermis, and lateral line; the bipotent vessel/artery and melanophore/xanthophore. After amputation, the labeled cells always regenerated the same cell type and did not contribute to other cell types [105]. Also, after amputation of caudal fins using fish with labeled cells in the body, but not in the fin, no labeled cells were found in the regenerate, excluding a distant cell contribution during regeneration [105].

This apparent variety of mechanisms (Table 2) recapitulates the heterogeneity of tissue repair and regeneration of vertebrates. Thus, a better understanding of blastema origin will have implications in regenerative medicine.

## 5. Epigenetic Changes in Regeneration

When damage or injury triggers appendage regeneration, a series of cellular and molecular responses are activated, such as inflammation, dedifferentiation, proliferation, and differentiation. Currently, a great effort has been placed in regenerative medicine to reprogram somatic cells toward a stem-like state, and then induce their differentiation to cells with the desired functionality. To this purpose, the understanding of the epigenetic changes regulating these processes is fundamental.

Epigenetic regulation of gene expression can be achieved through covalent modification of DNA and associated histones, chromatin remodeling, and the action of non-coding RNAs [108]. The best characterized is DNA methylation occurring on cytosines belonging to cytosine and guanine repetitive clusters (CpG islands) [109]. Methylation maintenance relies on DNA methyltransferase 1 (DNMT1), while *de novo* methylation requires DNA methyltransferase 3 (DNMT3) and results in a change of expression [109]. DNMT3s are essential during early embryonic development [110] and are important regulators of cell differentiation [111,112]. Recent studies indicate that *de novo* DNA methylation also acts during regeneration to control cell behavior and fate. A nerve-dependent downregulation of the axolotl DNMT3a during limb regeneration was associated with the transition of the wound epidermis into a functional AEC [113]. In an experimental assay, the treatment of axolotl skin wounds with a DNMT inhibitor induced cells to participate in blastema formation and resulted in a regenerative response [113]. After zebrafish caudal fin amputation, the level of cytosine methylation was transiently reduced and associated with an increased expression of demethylation genes [114]. The degree of methylation was gradually recovered during fin regeneration and at 72 h post-amputation the expression of three zebrafish orthologues of mammalian *DNMT3A* and *DNMT3B* (*dnmt3aa, dnmt3ab, and dnmt4*) [115,116] was clearly detected in blastema cells [117]. The presence of a dynamic role of DNA methylation has yet to be investigated during lizard tail regeneration. MicroRNAs (miRNAs), short non-coding RNAs, mediate post-transcriptional repression by binding the 3′ untranslated region (UTR) of target mRNAs. Among their multiple roles as epigenetic regulators, they are fundamental to control cell differentiation and fate [118].

Microarray analysis of miRNAs expression during axolotl limb regeneration reported the overexpression of miR-21 in the mid-bud blastema, which presumably maintains blastema cell differentiation through a negative regulation of the developmental gene *Jagged1* [119].

miR-133a, miR-133b, and miR-206, already known as regulators of muscle development and differentiation [120,121], were found to be differentially expressed during *Anolis carolinensis* tail regeneration, with an increased expression in the proximal portion with respect to tail tip [122].

miR-133 was found to be also highly expressed in uninjured fins and depleted during zebrafish caudal fin regeneration [123]. Interestingly, the depletion was dependent on FGF signaling, indicating a significant role of this pathway in maintaining low miR-133 levels besides its essential role for blastema formation and maintenance. It would be interesting to investigate its epigenetic role also in lizards. Another study on zebrafish caudal fin regeneration indicates that the other master regulator of regeneration, the Wnt/β-catenin pathway, is also regulated by miRNAs. This study showed that miR-203 specifically targets the Wnt signaling transcription factor Lef1 and acts as a negative regulator of regeneration [124].

These data indicate that epigenetic changes, as during development, play fundamental roles during regeneration and are sometimes conserved in different species. The success of regenerative therapies based on induced dedifferentiation will likely depend on the ability to recreate and regulate these specific epigenetic changes.

## 6. Patterning and Positional Memory

A classical regeneration restores not only the lost appendage, but also its three-dimensional architecture and size. Indeed regeneration, similar to development, involves morphogenetic processes to shape the regenerated tissues. The proximodistal, anteroposterior, and dorsoventral developmental axes are reorganized within the regenerative blastema, guided presumably by diffusible as well as positional clues permitting, among others, size preservation. One of the first and most popular explanations for this phenomenon was proposed by Wolpert. He hypothesized a simple coordinate system that provides cells with a positional value that, together with the cells’ genetic heritage, determines a specific response during ontogenetic pattern formation [125]. In the context of a regenerative process, this phenomenon is defined “positional memory” and refers to the information that blastema cells retain about their position along the proximodistal axis.

The properties of blastema during the patterning and outgrowth phase were initially studied using regenerating salamander limbs. Butler and colleagues proposed *the rule of distal transformation* that refers to the ability of blastema to regenerate along any level of the proximodistal axis and always proceeds in a direction distal to the stump surface. The rule was tested by inverting the proximodistal axis of an *Ambystoma* regenerating limb by suturing the autopod to the body trunk, and then cutting the circularized limb. This gave rise to two limbs, one of which had reversed polarity, but still regenerated all limb elements distal to the level of the amputation surface (Figure 4A) [126].

The stump tissues can also “sense” discontinuities along the proximodistal axis. Experiments in which gap discontinuities were created in regenerating limbs revealed that cells tend to proliferate and eliminate them in a process named intercalation. For instance, when a distal blastema was grafted onto an upper limb stump, the intermediate limb parts arose from the stump tissue, whereas in the converse experiment, grafting of a proximal blastema onto a wrist stump resulted in regeneration directly from the wrist and intercalation did not occur (Figure 4B) [127]. Much remains to be learned from a molecular and cellular point of view on how regenerating appendages are precisely replaced. Various observations suggest that positional memory may be related to a proximal-distal gradient of ligands and receptors with higher concentrations inducing more distal fates. For instance, one of the most intriguing findings was the ability of retinoic acid (RA) to reprogram proximo-distal values of blastema cells. Treatment of regenerating salamander limb with RA resulted in proximalization of the wrist blastema, so that an entire limb could be regenerated. The extent of proximalization was dependent on the exposure time and RA concentration [128]. A differential screening comparing newt RA-treated distal limb blastema with untreated one was used subsequently to identify cell-surface molecules able to act as effectors of positional memory [11]. This method led to the identification of Prod1, a GPI-anchored receptor belonging to the Ly6 family. The likely role of Prod1 is to determine positional memory via a gradient of cell adhesion. Indeed, the gene is highly expressed in newt proximal limb blastema and upregulated in response to RA [11,13]. Furthermore, overexpression of Prod1 in axolotl limb blastema re-specified cells to attain a more proximal position in the regenerated limb [129]. Lizard *Gekko japonicus* CD59 expression, originally interpreted as the ortholog of Prod1 [11], was found to increase following tail amputation and was upregulated by RA [12]. Also, overexpression of CD59 in lizard distal blastema re-specified transfected cells to a more proximal location, similarly to what has been shown during salamander limb regeneration [12]. Although no putative CD59 homolog has been identified in zebrafish, some CD59 related transcripts and proteins expressed in proximal to distal gradients have been found in the regenerating caudal fin. Furthermore, *aldh1a2* involved in RA synthesis and the anterior gradient protein *agr2*, identified as a Prod1 ligand [130], are expressed in proximally enriched gradients [41].

The 3D structure of the Prod1 protein was solved by nuclear magnetic resonance (NMR) spectroscopy in order to perform a comparative study to other members of the three-finger protein (TFP) superfamily, to which Prod1 belongs. The molecular phylogeny, based on both sequence and structural criteria, indicates that Prod1 has no known orthologues in other vertebrate taxa [131].

Patterning and size control during growth and regeneration are also regulated by bioelectric signaling. Several well-studied zebrafish mutants with mutations in K^+^ channels show altered appendage growth. The *another longfin* (*alf*) phenotype is due to a mutation in Kcnk5b, which directly increases potassium conductance, and causes fin overgrowth [132]. In contrast, the *short fin* (*sof*) mutation occurs in the *connexin43* (*cx43*) gene, which encodes a protein involved in gap junction formation, and results in shorter fins [133]. The correlation between ion current and patterning is largely unknown, and possibly depends on electrical coupling between cells [134,135]. It is interesting to notice that ion current has been implicated in several examples of tissue regeneration, including salamanders and newts. For instance, after amputation, forelimb stumps are traversed by a strong, steady, and polarized Na^+^ electric current that originates at the injury site [136]. Reducing this current resulted in inhibited or abnormal limb regeneration [137].

Recently, a role for heat shock proteins (HSPs) as regulators of shape during tail regeneration of the newt *Pleurodeles waltl* has been proposed [138]. Considering that HSPs expression has been proven necessary for blastema formation during axolotl limb [139] and zebrafish caudal fin regeneration [140,141], it would be of interest to elucidate if they have morphogenetic functions also during regeneration of other organisms.

Taken together, these findings suggest that mechanisms of patterning and positional memory could be conserved and adapted in the different species. Further studies are needed to clarify the effectors of the latest stages of regeneration and how they are phylogenetically correlated. Most importantly, it will be relevant to determine which of these signals have been further preserved in the evolution and could be potentially reactivated for biomedical purposes.

## 7. Role of Inflammation

The key role of macrophages as key mediators of inflammation during mammalian tissue repair associated with the deposition of fibrotic tissue and scar formation is well known [142]. In the past decades, a large body of data has been collected on the role of macrophages in organizing and controlling the inflammatory responses during wound healing of species characterized by high regenerative potency. Godwin et al. contributed to one of the major studies on urodele inflammatory response after limb amputation. The authors demonstrated an increased level of myeloid cells in regenerating axolotl tissues within one day after amputation. During the early stages of regeneration, a balanced expression of pro-inflammatory cytokines (TNF-α, IL-1β, IL-16, IL-17, IL-23, IFN-γ) and anti-inflammatory cytokines (IL-4, IL-5, IL-10, IL-13) is required. Indeed, the use of clodronate liposome, a well-established method to selectively deplete macrophages in vivo, led to an unbalanced increase of pro-inflammatory/anti-inflammatory cytokine ratio, reduced MMPs secretion, and resulted in non-functional wound epidermis [143]. The tracking of myeloid cells in transgenic zebrafish lines revealed a similar response, with circulating neutrophils and resident macrophages accumulating in the injury site at 6 h post-amputation and 3 days post-amputation, respectively. Although the creation of a neutropenic environment did not considerably affect the regeneration rate, genetic macrophage ablation resulted in reduced tissue regrowth in this model [53]. A recent study explored the role of macrophages in lizard regenerating tail, revealing an increase in the phagocytic cell population within the blastema starting from 3 days post-amputation. Clodronate liposome treated lizards failed to form blastema and to regrow amputated tail, further confirming the importance of macrophages in regeneration [144]. Another recent work reported the presence of anti-inflammatory M2-like macrophages underneath the wound epidermis and near the ependyma in the regenerating *Podarcis muralis* tail, which similarly to the amphibian limb, support and stimulate regeneration [145]. This is in contrast with lizard scarring limb in which only M1-like macrophages were detected [146].

Taken together, these results indicate that a timely activation of macrophages is required during the early steps of regeneration for a correct blastema formation and the onset of the regenerative process. During this phase, macrophages contribute to the establishment of an inflammatory environment through the secretion of cytokines and MMPs to degrade ECM components and tissue debris facilitating cell migration and communication [17,143,144]. Furthermore, the presence of an early anti-inflammatory response is related to a resolutive and anti-fibrotic activity that may be critical to both resolve inflammation and to carry on regeneration [147] by preventing fibrotic deposition and scarring [142]. The clearance of senescent cells is another suggested role for macrophages during the early regeneration phase. In fact, amputation or injury triggers apoptosis and cellular senescence [148,149]. In salamander for instance, it was demonstrated that senescent cells were transiently induced following amputation and subsequently cleared by macrophages. Since the depletion of macrophages led to senescent cells accumulation [150], perhaps the effective removal of senescent cells by them is a physiological mechanism during the regenerative program. Nevertheless, a role for macrophages during the later steps of regeneration cannot be excluded. Indeed, ablation of macrophages after blastema formation does not affect regeneration rate, but compromises fin patterning and bone formation in zebrafish [53]. In salamander, the lack of macrophages during the later stages of regeneration associates with a reduction of surface vasculature, implying another possible functional role for macrophages during tissue outgrowth [143].

## 8. Immune System and Regeneration

The strong, scar-free regenerative potential in selected vertebrate species has raised the question whether differences in immune systems can affect the outcome of tissue regeneration. One of the indications of a negative effect of the immune system on regeneration was described in anuran amphibians which lose their robust regeneration ability after metamorphosis when the immune response is enhanced [17].

Lizard, salamander, and zebrafish heavily rely on an efficient innate immunity, consisting of diverse components such as antimicrobial peptides, neutrophils, macrophages, and the complement system [151,152,153,154]. Their adaptive immunity, however, differs from that of mammals by missing lymphoid organs such as Peyer’s patches and lymph nodes [151,155,156].

Lizards, of the three species the closest related to mammals, share similarities with them in terms of generation of antibody diversity, but produce slower and less robust adaptive humoral responses [151]. Their B cells are characterized by phagocytic activity, a feature shared with salamanders and still under debate for zebrafish [155,157,158], and act as a component of innate immunity [151]. The strong innate immune system seen in this group of vertebrates could be sufficient to protect from microbial infections present in the first stages of life, making efficient adaptive immunity non-essential.

Salamanders are generally considered “immunodeficient” since they have a weak cell-mediated immune response, and their humoral immunity is based only on IgM production and is apparently amnestic [152]. Moreover, they have a restricted MHC class II diversity, likely resulting in a poor T-helper stimulation and a low cytokine synthesis [159] that could in part explain the low inflammatory response seen during limb regeneration.

Zebrafish develop their adaptive immunity later, at the end of their embryo phase [160]. Their T-cell responses are negatively affected by low body temperature [161]. RAG1 protein, essential for antibody and T-cell receptor V(D)J recombination, is conserved in zebrafish. However, homozygous *rag1* mutant fish, lacking functional T and B lymphocytes, reach adulthood and are fertile [162,163], suggesting that these cells are likely not necessary for an efficient immune response.

An ultrastructural study of the regenerating tail of the lizard *Podarcis muralis* showed that repeated amputations or cauterization trigger an increase in the number of immune cells such as granulocytes, macrophages, and lymphocytes within the blastema and led to the deposition of fibrinoid material with scarring [64]. The study supports the hypothesis that the evolution of the immune system may be associated to the reduced regeneration ability in lizards, and in amniotes in general. Further studies are needed to clarify how immune cells and mesenchymal cells of the blastema communicate and interact during regeneration.

## 9. Nerve-Dependency of Regeneration

The knowledge that the nervous system is involved in the regenerative process was reported as early as 1823, when Todd demonstrated that denervation of the amputated salamander limb inhibited or caused defective regeneration [164]. Nerve-dependency became one of the most intensively studied topics in regeneration only from the mid of the 20th century when Singer performed a series of elegant experiments on denervated salamander limbs (reviews of his work [165,166]). He pointed out that not a specific type of nerve, but all nerve fibers contribute equally to the regenerative process. Also, he observed that regeneration never occurred if the number of nerves did not exceed a certain threshold [167,168]. Based on these findings, the neurotrophic hypothesis was formulated, stating that the nerve produced trophic factors crucial for the survival and proliferation of blastema cells. Regenerating nerves interact with the wound epidermis and sustain apical epidermal cap (AEC) formation and maintenance [169]. Potential trophic factors include substance P, insulin, transferrin, neuregulin-1, FGFs and BMPs [170] but the main candidate trophic factor is the anterior gradient protein (AGP), a ligand for the blastema cell surface receptor Prod1 cited previously, that offers a molecular explanation for nerve-dependence in urodeles. AGP is strongly expressed in Schwann cells of newt limbs in the first phase of regeneration and later shifts to the glandular structures of the wound epidermis [130]. Similarly, in axolotl, which do not possess subepidermal glandular cells, AGP expression was observed in Leydig cells [171], and denervation of limbs abolished AGP production. This suggests that, in both newt and salamanders, nerves induce the production of the protein that is then secreted by epithelial cells to blastema cells with a holocrine mechanism.

Nerve-dependency of regeneration is a phenomenon that extends across phylogeny. Destruction of the spinal cord disrupts regeneration in lizard in a mechanism following the concept of a required threshold of nerve fibers [172]. In particular, the ependymal tube seems to be crucial for initiating regeneration [173]. If white and grey matter are damaged alone, but the ependymal population is maintained, regeneration still occurs [174]. Furthermore, transplanted segments of the ependymal tube can initiate blastema formation and produce ectopic tails at grafting sites [174,175], demonstrating that this structure alone is capable of starting regeneration. How exactly the ependymal tube mediates regeneration remains unclear.

The supporting role of nerves to wound epidermis and AEC is confirmed in zebrafish regenerating pectoral fin. In this model, thickening of the wound epidermis and AEC formation are negatively affected by denervation [176]. The demand for regeneration of a threshold number of nerve fibers is extended also to this organism. Indeed, the complete absence of nerve fibers impedes blastema formation and regeneration, whereas a reduced number of nerves allows smaller and deformed fin to regenerate [176]. The AGP homolog *agr2* was found to be expressed in the mucus-secreting cells of the epidermis, although no difference was seen between control and denervated fins. Furthermore, if denervation was applied after blastema formation, fin rays regenerated, but presented a narrowed and crooked structure [176]. This last observation supports the critical role of nerves during the initial stages of regeneration and points to their effect also during regenerative outgrowth and patterning.

An exception to the nerve-dependency phenomenon is the aneurogenic salamander limb created by removing the neural tube from embryos. Aneurogenic limbs have never been innervated but are still able to regenerate [177]. However, aneurogenic limbs become nerve-dependent upon complete innervation from the host [178]. Thus, a fascinating explanation for why aneurogenic limbs regenerate is that critical regeneration factors are initially produced in the developing limb, but after innervation, they are supplied by nerves. After that, a neural–epidermal relationship is created, and the epidermis becomes dependent on neural factors to promote and sustain the regenerative process [170,179]. It will be interesting to investigate the conservation of this phenomenon in other species.

Taken together, the findings reported in the different species support the idea that nerves produce trophic factors promoting and regulating blastema proliferation and differentiation and possibly the subsequent patterning. The nerve-dependent regeneration is detected in most regenerating organisms, but mechanisms differ. Investigating the evolution of nerve-dependency in different regenerating vertebrate species would likely help to get further insight into this process.

## 10. Molecular Pathways Implicated in Regeneration

Further identification of the signaling networks involved in the regeneration remains one of the key research directions in this field. In the past decades, much knowledge was acquired about the molecular mechanisms that mediate the regenerative processes in various organisms, pointing to the presence of shared signals (Figure 5). Indeed, the molecular comparison among different regenerating vertebrates reveals that FGF and Wnt/β-catenin signaling pathways play fundamental roles during blastema formation and proliferation in all of them and can reasonably be considered as universal regulators of regeneration.

An interesting study comparing lizard regenerating tail and scarring limb transcriptomes shows that many genes belonging to the Wnt pathway (*wnt2b*, *wnt5a*, *wnt5b*, *wnt6*) are upregulated in the tail blastema, but absent in the scarring limb. Instead, the scarring limb expressed genes that act like inhibitors of the Wnt pathway (such as *dkk2),* impeding cell proliferation, stimulating differentiation and tissue repair leading to scarring [21]. Wnt pathway is also required for axolotl limb regeneration. Overexpression of Axin1, an intracellular inhibitor of the Wnt pathway, in the regenerating limb inhibited regeneration and resulted in spike-like regenerated limbs lacking the digits [180].

FGF1 and FGF2 were detected in the regenerating lizard tail, especially in the wound epidermis and in the spinal cord [181] and inhibition of their activity eventually stopped regeneration [182]. Interestingly, FGFs are absent from lizard scarring limb [183] and if administrated after lizard limb amputation stimulate the formation of the AEP and the regeneration of cartilaginous femur, tibia, and fibula, although never of an autopodium (digits), confirming their requirement for a successful regeneration [184]. During salamander limb blastema formation, *fgf8* and *fgf10*, already present in the intact limb, are upregulated [185]. It has been proposed that this pattern of expression correlates with the “readiness” to regenerate [185]. Denervation of the limb prevents *fgf8* and *fgf10* upregulation [185], given their neurotrophic properties [186].

In zebrafish, the wound healing phase begins immediately following the amputation coinciding with the induction of *wnt10a* and *fgf20a* [43,56]. Disruption of either signaling pathways using transgenic fish lines or pharmacological inhibitors resulted in an abnormal wound epidermis and altered blastema formation [43,56,187]. A strong expression of *aldh1a2*, the rate-limiting enzyme in RA synthesis, was also detected early in the stump. RA administration increased the levels of Wnt/β-catenin and FGF ligands *wnt10b* and *fgf20a*, demonstrating that these signals belong to a complex network orchestrating and coordinating blastema formation [188]. After blastema forms, it compartmentalizes in two domains: a low proliferating distal-most blastema, and a large proximal blastema populated by highly proliferating and differentiating cells [81]. RA, Wnt/ β-catenin and FGF pathways remain active in the blastema to sustain proliferation and survival. Non-canonical Wnt signaling mediated by *wnt5b*, instead, plays an antagonistic role and inhibits regeneration [43].

The induction of cell proliferation is one of the essential and early stages of regeneration. Recently, Myristoylated alanine-rich C-kinase substrate (MARCKS)-like protein (MLP) was identified to be responsible for the induction of an early regenerative response in axolotl tail and limbs [189]. MARCKS, and MARCKS-like proteins, have been previously described as widely distributed, intracellular substrate for protein kinase C (PKC) that coordinate membrane–cytoskeletal signaling events, such as cell adhesion, membrane trafficking, and phagocytosis [190,191]. In vivo injection of Axolotl MLP (AxMLP) into uninjured axolotl tissue induces cell cycle re-entry of multiple cell types, such as epidermis, spinal cord, notochord and myotubes, while its morpholino knock-down prevents cell cycle onset and regeneration [189]. Interestingly, despite the protein lacking a signal peptide [192], immunohistochemical analysis of AxMLP distribution in epidermal and spinal cord tissues shows that the protein was mostly cytoplasmic in uninjured tissue, and, following injury, was translocated to the membrane and secreted [189]. How AxMLP induces cell proliferation and is extracellular release is still unknown. The induction of MARCK-like protein in the lizard regenerating tail suggests that this protein may be an initiator of regeneration also in this organism [21,70]. Since MLP secretion was reported only in axolotl so far, it would be interesting to assess if this unconventional process is also present in other vertebrates with regenerative ability.

The communication between wound epidermis and blastema is fundamental to achieve regeneration. IGF signaling, already known as a regulator of cell growth during development [193], seems to mediate this epidermis-blastema interaction [46]. The ligand *igf2b* is expressed in the zebrafish caudal fin blastema, whereas signal-responsive cells expressing its receptors *igf1ra/b* are located in the wound epidermis. The activation of Igf1r signaling in the wound epidermis promotes and sustains cell survival and wound epidermis specification, but more importantly, it is required also for the normal blastema formation. The inhibition of the signaling affects the expression of both wound epithelial markers *wnt5b* and *lef1* and blastemal markers *msxb* and *fgf20a* [46].

Whereas other members of the TGF-β superfamily are needed in the early phases of the regenerative program, such as activin-βA [99,194], the bone morphogenetic proteins (BMP) pathway is recruited later in the post-blastema phase to promote patterning and morphogenesis. As revealed by investigation on lizard *A. carolinensis*, BMP signaling mediated by BMP-2 and BMP-6 regulates proximal cartilage proliferation and hypertrophy during tail regeneration [100].

Dedifferentiation of muscle cells in the salamander limb blastema in vivo was shown to occur thanks to the activation of BMPs [195]. Another proposed role for BMP signaling during limb regeneration is to trigger cell condensation and apoptosis. In fact, the overexpression of Noggin, a general inhibitor of all BMPs, inhibits both processes [196].

BMPs are expressed in fin osteoblasts during regeneration in zebrafish [197]. A recent work suggests that BMP signaling acts in the later phases of regeneration by activating *dkk* proteins to counter growth induced by the Wnt pathway and promote osteoblasts differentiation [198]. The overexpression of chordin, another BMP signaling inhibitor, in the regenerating caudal fin leads to a reduction in bone matrix deposition due to defects in the maturation and function of bone-secreting cells [197]. *Hsp70* promoter-driven expression of *noggin 3* in transgenic fish causes a regression of the outgrowth in regenerating caudal fin exposed at heat shock between 3 and 4 days post-amputation (dpa) [199].

*Sox9a* and *sox9b* genes, known to be involved in condensation and chondrocyte differentiation, are detectable in zebrafish regenerating fin, which has a dermal origin. Their different pattern of expression (mesenchymal cells of the blastema and basal epidermal layer, respectively) and different response to BMP inhibition (only *sox9a* is downregulated) suggest that they could be regulated by different pathways in the regenerate [197].

Msx homeobox genes, the immediate early BMP response transcription factors [200], are thought to have a role in development and regeneration as well. Despite the lack of expression in *Anolis carolinensis* [20] and *Gekko japonicus* [201], a proximal–distal gradient of Msx1-and -2 generated from the apical blastema was detected in the regenerating tail of the lizard *Podarcis muralis* [202], suggesting a role in blastema maintenance for these proteins in this model. In regenerating urodele limbs, the two Msx genes are differentially expressed: the Msx2 gene is rapidly induced in the wound epidermis and subjacent tissues after amputation, while the Msx1 gene expression is induced only in blastema cells [203]. They possibly have different functional roles, with Msx2 being implicated in wound epidermis-blastema interaction and Msx1 acting to control blastema cells proliferation. The induction of Msx1 expression has been demonstrated to activate salamander myotubes cell cycle entry in vitro [107], suggesting that this molecule may control cellular plasticity during salamander limb regeneration. Finally, Msx genes also have been demonstrated to participate in zebrafish fin regeneration [204]. Zebrafish *msxb/msxc* genes, related to Msx genes of tetrapods [205], are upregulated early in the blastema and were demonstrated to be dependent on Fgfr1 expression [187]. Also, *msxb* shows an interesting pattern of expression during the regenerative outgrowth. In the immature blastema, *msxb* is expressed by slow proliferating blastemal cells. After blastema compartmentalization, *msxb* expression localizes in the distal blastema, suggesting that it acts in this compartment to regulate cell cycle and proliferation [206], similarly to axolotl Msx1 [203]. Confirming this functional role, morpholino knockdown of *msxb* prevented cell proliferation and fin outgrowth [207].

The Hedgehog signaling is possibly the key player of the last phase of regeneration. The morphogen sonic hedgehog (Shh) is expressed in the regions surrounding the ependymal tube in the regenerating lizard tail, where it establishes proximal organization centers and induces chondrogenesis in lizard blastemal cells. Its pattern of expression along the circumference of the tail is probably related to the characteristic lack of dorsal-ventral patterning of the regenerating tail [100]. Developmental studies demonstrated that outgrowth and patterning of salamander limb bud is coordinated by a positive feedback loop between Shh and Fgf [208]. In salamander regenerating limbs, *shh* localizes in the posterior part of the blastema, immediately adjacent to the AEC [209]. A recent study demonstrated that stimulating the Shh signaling pathway in anterior innervated limb tissue was sufficient to sustain limb regeneration and form a normal limb. The posterior cells, instead, are “refractory” and do not respond to *shh* stimulation but proliferate only after overexpression of *fgf8*. Thus, similarly to the developmental process, posteriorly localized *shh* signal supports anterior expression of *fgf8* in a positive feedback loop that ultimately leads to proliferation and eventually patterning of the limb [210].

In zebrafish regenerating fin, the *shh* gene is expressed in the cells of the basal layer of the epidermis, in the area surrounding newly forming fin rays [211]. The bone morphogenetic protein *bmp2* is expressed in both the basal epidermal cells and the adjacent blastema. *Shh* and *bmp2* expression in epidermis is further regulated by cell-cell interactions between basal epidermal cells and blastema [211]. Ectopic expression of *shh* or *bmp2* in the blastema results in bone deposition between fin rays and mispatterning of the regenerated fin [212]. The exposure to cyclopamine, inhibitors of the Shh signaling, reduces proliferation and differentiation of specialized bone-secreting cells within the blastema [212]. *Shh* has also a role in bone patterning and ray branching morphogenesis [213,214].

Taken together, these findings show to what extent the signaling pathways underlying the regenerative processes in different vertebrates are shared. Despite some difference, it seems that there are conserved molecular key players of regeneration that induce similar cellular responses and similar outcomes across phylogeny. Our knowledge is far from complete, but future studies will help unravelling what are the precise functions of these signals and how they differ among different organisms and perhaps will allow for the molecular manipulation of the regeneration process in mammals.

## 11. Regeneration and Aging

The aging process is associated with an imbalance between the accumulation of senescent cells and the rate of cell renewal [215]. Interestingly, many invertebrates with high regenerative potential as adults, such as hydra and planaria, show no signs of aging [216,217,218] supporting a link between regeneration and aging. Various intrinsic and extrinsic factors are associated with the decline of regenerative capacity upon aging: genomic instability, telomere shortening, oxidative damage among the others [219]. The comparative studies of vertebrates with extensive regenerative abilities is a powerful tool to better understand the relationship between regeneration and aging. Unfortunately, only limited work has been done on the topic, but the consensus is that embryos, larvae, and juveniles of diverse species have a stronger regenerative potential compared to older individuals. A study on embryonic and post-natal common lizards (*Lacerta vivipara*) demonstrated that after tail amputation at different stages of embryonic development, regeneration occurs only in late-stage embryos, while early-stage embryos fail to regenerate [220]. The explanation for these unexpected results could be linked to the evolution of tail regeneration in lizards. The process of autotomy as an anti-predation strategy is probably the consequence of natural selection and has no meaning in amniote embryos, which are already protected by their mother’s body. Further studies are needed to confirm this hypothesis.

Salamanders and zebrafish, however, exhibit extensive and time stable ability to regenerate their appendages. It was demonstrated that salamanders have a strong immune-mediated mechanism that rapidly recognizes and clears senescent cells during limb regeneration [150]. Moreover, both their normal and regenerating tissues show no accumulation of senescent cells with aging [150]. Taken together, these findings may indicate that efficient turnover of senescent cells is a physiological mechanism that can mediate aging and regeneration in these organisms. Thyroxine-induced metamorphosis in age-matched axolotls resulted in reduced regeneration rate and fidelity and increases in the occurrence of morphological alterations in the limb [221], indicating patterning and growth alterations. Also, metamorphosis seems to alter cell dynamics during regeneration and decreases the number of proliferating blastema cells [221]. Metamorphosis is not necessarily related to age; however, these findings imply that developmental stages are important modulators of regeneration rate. Another view of metamorphosis as a destructive and reconstructive process providing molecular and cellular sources that can be re-utilized during adult life to regenerate appendages has been recently suggested [222,223]. Indeed, various genes expressed during metamorphosis are also upregulated during regeneration in amphibians and fish [223]. More detailed research will be relevant to clarify this correlation.

The relationship between aging and regeneration rate and ability is poorly understood also in zebrafish. Although age and sex-related differences seem to affect regeneration of the pectoral fin [224], the results reported for caudal fin regeneration remain controversial. A study showed that fin regeneration was impaired, and telomerase activity was reduced in the regenerating fin of older fish, indicating a correlation between telomerase activity, a marker of aging, and regeneration efficiency [225]. Genotoxic stress caused by ionizing radiation was also shown to enhance the process of aging and impair caudal fin regeneration [226]. Contrary to these results, another study reported a comparable caudal fin regeneration rate between young and old fish and supported the idea that zebrafish keep their regeneration capacity throughout the lifespan [227].

Many factors affect the aging and regenerative processes and unraveling the correlation among them is difficult. More studies are needed in order to clarify what are the regenerative-time windows in vertebrates and the nature of the genetic and epigenetic alterations that occur during lifespan and change regenerative capacity. This type of study would have implication not only in regenerative medicine but could also help to understand what are the factors that affect aging and age-related disorders in humans.

## 12. Regeneration Models as a Tool for Biomedical Research

Zebrafish is not only a fascinating example of vertebrate regeneration, but also represents a well-known vertebrate model for disease and has already contributed to several examples of successful phenotype-based drug discovery, providing a tool with many different applications in biomedical research, especially in the field of bone biology. Indeed, zebrafish skeleton conserves the general basis of development, gene expression and cell types that are found in mammals [47]. In this scenario, the remarkably fast regeneration of zebrafish caudal fin provides an excellent tool to easy monitor bone formation over time in healthy and pathological conditions [47]. The evaluation of bone regeneration rate can be performed in combination with drug administration to discover new molecules promoting regenerative and/or osteogenic activity. As an example, after fin amputation bone growth can be followed in juveniles kept in small tanks with limited water volumes and amount of drug, allowing high throughput analysis at low cost [228]. To these purposes, the availability of transgenic lines and bone assays including fluorescent dyes such as calcein and alizarin red, which stain calcified tissue, provides a simple and fast way to visualize the effect of a specific compound on bone [229]. One of the challenges for such approaches is how to control and predict the amount of the administrated drug although direct injection of the drug in fin rays can be used to overcome this limitation [230].

At the time, regeneration in other vertebrate models such as lizard and salamander is not exploited for drug discovery, but the rapid advances in molecular techniques could likely overcome such obstacle.

## 13. Conclusions

Mammals, humans included, are incapable of regenerating amputated or lost limbs. Damage or injury can be sometimes life threatening and is reasonable to wonder how we might be able to enhance our own regenerative potential. For this purpose, the understanding of cellular and molecular mechanisms orchestrating regeneration is crucial. In the past years, the field of regenerative biology has seen considerable progress, particularly thanks to the study of organisms with a high regenerative potential. Although morphological mechanisms and cell contributors of regeneration can vary widely even among close species, the comparison of the three vertebrate models here discussed allows to identify an overall common strategy to successfully restore a lost appendage. So, what makes this possible? As it seems, a combination of multiple factors. First, the formation of a thick wound epidermis/AEC that supports the regenerative process without scarring. The formation of a blastema, which provides the source of differentiated cells that will restore the lost tissues is the key process present in vertebrates with regenerative ability. Proper activation of specific molecular pathways has been proven necessary for successful regeneration. The data collected on the three described models unequivocally points to the role of Wnt/β-catenin and FGFs as master regulators of the process. The expression of these key molecules is not only necessary for regeneration to occur but is also able to promote it in organs with limited regenerative potential, such as lizard limb. Blastema growth is also strictly related to the level of immunosurveillance. Lizard tail, salamander limb and zebrafish caudal fin can be considered immuno-privileged organs, in which inflammation triggered by injury equilibrates with the presence of healing, anti-inflammatory macrophages.

Each model presents advantages and limitations. Salamanders are considered the master regenerators being the only vertebrate able to regenerate a full limb. Since its discovery in the 18th century [231], the regenerating limb system was used to perform fundamental experiments to delineate the basic properties of regeneration. However, many aspects of the urodelian life cycle are at odds with those of higher vertebrates. The axolotl, the most used salamander for regenerative studies, is neotenic and retains all its juvenile features even when it reaches adulthood. Lizards, on the other hand, follow a similar development as mammals and therefore are more appealing from a developmental point of view. However, as amniotes, their regenerative ability is restricted to the complex structure of the tail, and their lost appendages undergo an imperfect regeneration.

Given the simplicity of using powerful genetic tools, zebrafish has quickly emerged as a model of choice to study regeneration. Zebrafish present several advantages, such as the external fertilization and the transparency of embryos that makes them particularly appealing for forward genetic approaches allowing to investigate regeneration in health and disease. 

Our knowledge of regenerative mechanisms is strictly related to the tools we can use. Thus, the possibility to apply more advanced genetic tools also on powerful regenerators such as salamander and lizard could represent a key step forward. The recent reports of Iberian ribbed newt (*Pleurodeles waltl)* and axolotl (*Ambystoma mexicanum*) complete genomes, together with the recent generation of a CRISPR/Cas9 Anolis lizard, will hopefully provide new tools to increase the use of these organisms to find the answer to regenerative questions still unsolved.

No unique model for regeneration exists, and the comparative study of vertebrates with high regenerative ability is fundamental to achieve significant insights on what are the factors driving regeneration, aiming to translate them into the field of regenerative medicine. Indeed, the field of regenerative biology has made an enormous step forward in the past decade, taking advance of imaging, genomics, and genome editing to identify key cell types and molecules involved in the regeneration of many model organisms. Yet, it can be difficult to foresee when and how findings from these studies will really advance regenerative medicine. The identification of genes modulating the origins and fates of blastemal progenitor cells will likely be the key to achieve appendage regeneration, and it will provide novel targets for gene manipulation in mammals.

## Figures and Tables

**Figure 1 cells-10-00242-f001:**
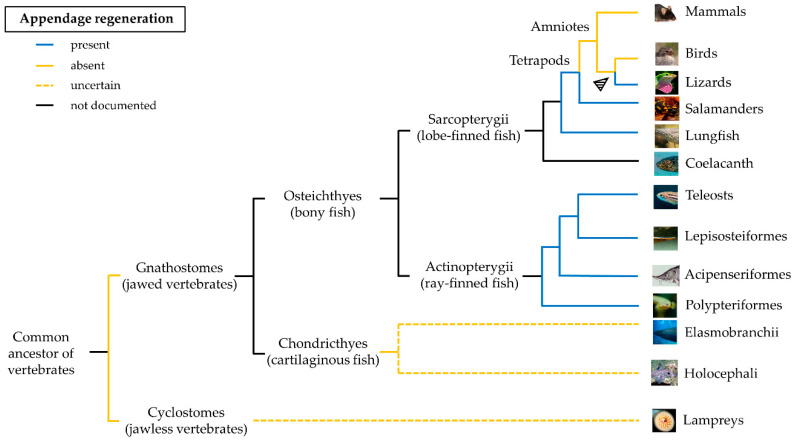
**Phylogenetic distribution of appendage regeneration among vertebrates according to the current literature.** The blue line indicates lineages containing one or more species capable of appendage regeneration, the orange line those incapables of regeneration, and the black line those in which appendage regeneration has not yet been documented. The dashed yellow line indicates the uncertain presence of appendage regeneration in chondrichthyes and cyclostomes. The arrow indicates the reactivation of the regenerative process in amniotes.

**Figure 2 cells-10-00242-f002:**
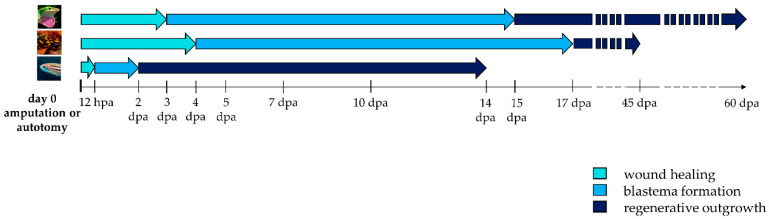
**Timeline activation of wound healing, blastema formation and regenerative outgrowth during appendage regeneration in vertebrates.** Arrows indicate the duration of each regenerative step in lizard, salamander, and zebrafish. Hpa: hours post-amputation; dpa: days post-amputation.

**Figure 3 cells-10-00242-f003:**
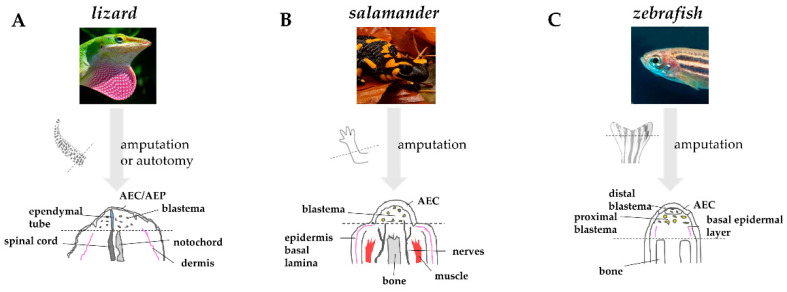
**Comparative anatomy of regenerating appendages in lizard, salamander, and zebrafish.** (**A**) Amputation or autotomy of lizard tail induces the formation of a cone-like shaped blastema, covered by a thick wound epidermis that will form the apical epidermal peg(AEP), whose correspondence to the apical epidermal cap (AEC) found in amphibians and fish is still uncertain. During blastema formation, likely in response to AEP signals, the central canal of the original spinal cord, the ependymal tube, elongates and infiltrates the proliferating tail blastema. (**B**) After amputation of salamander limb, the wound epidermis quickly covers the stump. Within days, the wound epithelium becomes innervated, thickens, and becomes a specialized AEC. The AEC induces dedifferentiation in the underlying stump tissue and attracts cells, which accumulate below the AEC to form the blastema. Modified from Payzin-Dogru and Whited, 2018. (**C**) After zebrafish fin amputation, epithelial cells migrate to cover the wound, forming the AEC. Under the AEC, stump tissues dedifferentiate to form the blastema. Within 24 h, blastemal cells segregate into two compartments: the distal blastema, populated by slowly proliferating cells, and the proximal blastema, in which cells rapidly proliferate and differentiate to replace the amputated tissue. Dotted lines indicate the site of amputation/autotomy.

**Figure 4 cells-10-00242-f004:**
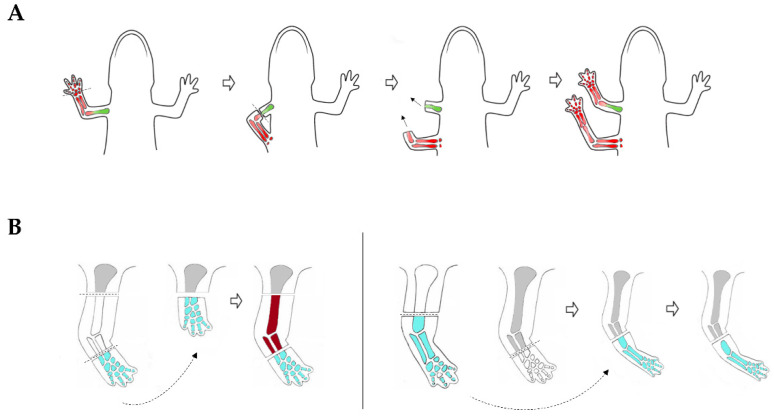
**Experiments illustrating the principles of positional memory in salamander limb.** (**A**) Rule of distal transformation. After the creation of a circularized limb, successive amputation induces the creation of two stumps, one with correct polarity and one with reversed polarity. Both stumps regenerate distal elements from the level of amputation, thus duplicating the distal segments already present in the reversed stump. The proximo-distal axis is indicated with a gradient of green (indicating proximal elements) and red (indicating distal elements). (**B**) Intercalation. Intercalary regeneration (dark red) occurs if a hand is grafted to an upper arm (left), but not if an upper arm segment is grafted to a forearm (right). Adapted from Carlson 2007.

**Figure 5 cells-10-00242-f005:**
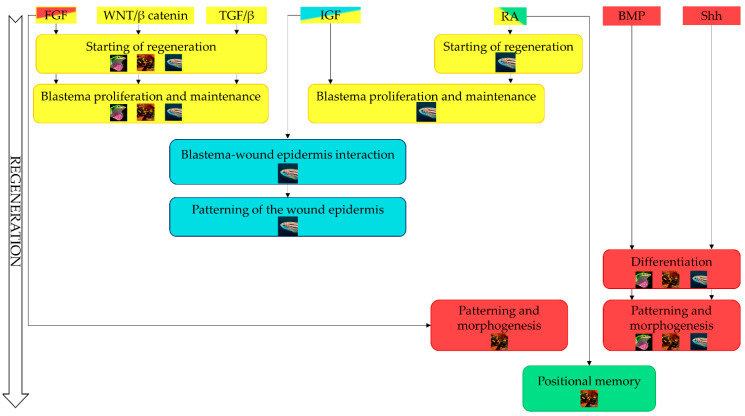
**Schematic representation of the molecular signaling regulating the steps of appendage regeneration in vertebrates**.

**Table 1 cells-10-00242-t001:** Experimental accessibility and tools for vertebrate species used in regeneration research.

	Lizard				Salamander		Zebrafish
	*A. carolinensis*	*E. macularius*	*G. japonicus*	*P. muralis*	*A. mexicanum*	*P. waltl*	*D. rerio*
**Experimental** ** Accessibility**							
Genome Size	1.78 Gb	2.23 Gb	2.55 Gb	1.51 Gb	~32 Gb	~20 Gb	1.67 Gb
Fertilization	internal	internal	internal	internal	external	external	external
Egg Size	~6 mm	~12 mm [48]	~13 mm [49]	~11 mm [50]	~2 mm	~2 mm	<1 mm [51]
Egg Transparency	no	no	no	no	yes	yes	yes
**Experimental Tools**							
Genome Annotation	Ensembl NCBI	NCBI	NCBI	Ensembl NCBI	NCBI	NCBI	Ensembl NCBI
Transcriptome Annotation	NCBI	NCBI	NCBI	NCBI	NCBI	NCBI	NCBI
Knock-down *****	no	no	no	no	yes [31]	no	yes [46]
Transgenesis	no	no	no	no	yes [30,38]	yes [52]	yes [43,53]
Knock-out/Knock-in	no	no	no	no	yes [54] **	yes [55] **	yes [42,56] ** #

* = morpholino ** = CRISPR/Cas9 # = ENU mutagenesis.

**Table 2 cells-10-00242-t002:** Mechanisms of cellular derivation of blastema in vertebrates. The green tick indicates what cellular conversion process has been documented during blastema formation in different species.

	Lizard			Salamander		Zebrafish
	*A. carolinensis*	*L. lugubris*	*P. muralis*	*A. mexicanum*	*N. viridescens*	*D. rerio*
Dedifferentiation		✔ [95]	✔ [94]	✔ [30,107]	✔ [102]	✔ [44,104,105]
Transdifferentiation		✔ [95]				
Stem cell recruitment	✔ [20,90,93]		✔ [89]	✔ [101,102]		

## Data Availability

Data sharing not applicable.

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
