# Peer review of "Appendage Regeneration in Vertebrates: What Makes This Possible?"

_cells, 2021, doi:10.3390/cells10020242_

Round 1
Reviewer 1 Report
This article is a valuable and meaningful review of research on the field of regeneration of appendages in lower vertebrates. Using three animal models (lizard, salamander, and fish) authors analyzed the similarities and differences in cellular and molecular mechanisms of the process. The review has been submitted for publication to the section: Stem Cells. The comments and remarks on this paper are below.
- As it was noted in the introduction (lines 59-65), in the review the authors discussed morphological, cellular and molecular aspects across the regenerative process in lizard, salamanders and fish and investigated differences that could have led to the loss of regenerative capacity in mammals. In this case, the authors’ choice of the section “Stem cells” is not quite clear and seems not quite correct. The review did not collect much information on the regeneration cell sources themselves, which can be, indeed, stem cells, stem-like cells, and differentiated cells of the stump participated in the restoration of the structures. In the review there is not enough information about their nature and biology, their progenitor traits and the stemness.
- It is also unclear why authors compare tail regeneration in the lizard and caudal fin in zebrafish with limb regeneration in salamanders. It seems much more reasonable to make comparison with tail regeneration in salamanders. Perhaps, that was due to the availability of much more information on limb regeneration then tail regeneration in Urodelean amphibians. However, this choice makes it difficult to discuss the data for the identification of main patterns. The authors themselves note (lines 209-210) that «…. contradictory findings could, in part, be explained by the embryonic differences between tail and limb buds, with the former having a pluripotent nature and being able to give origin to all the three germ layers”. Although both the tail and limb are the appendages these structure (apart from differences in regulatory mechanisms of their development) have significant and principal anatomical differences, such as the presence of the spinal cord and its involvement in the process of tail regeneration.
- Regeneration is most often viewed against the background of and alongside the development, often reproducing the molecular mechanisms of the latter. In the review such a link is not obvious and is not emphasized. There is not enough information about the expression of “developmental” transcription factors, as well as epigenetic changes, controlling the return of cells to dedifferentiated state in the blastema. This information can be given either in the chapter 4 “Role and origin of blastema: dedifferentiation versus stem cells” or separately.
- The data in the table 2 require a reference to the paper sources where given cell differentiation changes are documented. In addition I have to note that the terminology used in the literature with respect to the plasticity of cellular phenotypes participating in the formation of the blastema is not often precise enough. On another hand, short discussion of these cellular mechanisms (de- and transdifferentiation or growth of differentiated cells) would be one more powerful argument for publishing the review in the section “Stem cells”.
- Сhapter 5 gives the information which is emerging currently and shed light on the morphogenesis of the appendages regenerating from an already formed body part. Appendages regenerating by this way have the right structure and function, as it happens in the development and this point is important. Besides data presented in the chapter recently there are some results concerned the mechanisms of the establishing of regenerating tail form in Urodela (see: Radugina, Grigoryan, 2018).
- Chapter 9 “Molecular pathways implicated in regeneration” provides sufficient information where a significant part is devoted to FGF and Wnt/β-catenin signaling pathways. It has a useful and simple illustration (Figure 5) of the application points of these and some other signal molecules at different stages of the regeneration process. However the legend to Fig.5 needs one correction. It is written “Schematic representation of the molecular signaling network regulating the steps of appendage regeneration in vertebrates” but the network as such is not shown in the figure, the lines show only the ratios of signaling pathways with the stage-targets of the regeneration process.
- Chapter 10 “Regeneration and aging” could include a discussion of repeated regeneration, such as that described for the limb and lens of the eye in Urodela. In the latter case contrary to the belief that regeneration becomes less efficient with time or repetition, repeated regeneration, even at old age, does not alter newt regenerative capacity (see: Eguchi et al., 2011; Sousounis et al., 2015).
- The inscriptions on figures 2 and 3 are too small to read.
- The advantage of the review is the discussion of the question of accessibility to modern researches of known models appendage regeneration. A useful table showing the current state of molecular and genetic approaches for studying regeneration in lower vertebrates is provided.
Therefore, the paper of Valentina Daponte, Przemko Tylzanowski, and Antonella Forlino is original, modern, summarizing of a broad range of current issues in regeneration in lower vertebrates (185 cites). This makes the review useful and interesting for many readers. Nevertheless, it seems necessary to clarify the above points and to amend the text somehow where it is needed.
Reviewer 2 Report
General
The present Review presents an updated discussion on a number of aspects frequently treated in numerous papers dealing with the regeneration in vertebrates. Despite of this, the Review also deals with some interesting and original aspects such as the consideration of lizards as an important model for regenerative studies in addition to amphibian and fish models. However, in this regard the present Review misses to consider several old and more recent studies and papers on lizards regeneration that have analyzed in deep the influence of growth factors, dedifferentiation, stem cells, blastema formation, role of the immune system, genes, to name a few. Another point that should be addressed by the authors is the need to specify which vertebrate they are referring to in numerous sentences of their MS since the discussion often jumps among different vertebrates and the reader is not sure whether the reported data are referring to a fish or amphibians or lizards. In numerous cases the discussion moves from a species to another without specification and a logical sequence, so what is valid for one species may not be true for another. Since the Review underlines also lizard regeneration, I have provided some other references and indications to these studies, easily accessible from the internet, that the present authors may or may not consider (this is NOT mandatory for the final decision on the MS) but that surely will provide some further infos and updating on what is presently known in these reptiles, the closest models to mammalian regeneration. The Review should also terminate with more specific sentences addressing the final part of the title …: what make this possible? .I believe that some effort to address the indicated issues and re-organized part of the MS would definitely improve and attract attention on the present, interesting Review.
Specific
Title:….what makes this possible? :UNFORTUNATELY, IN THE PRESENT FORM, THE MS DOES NOT INDICATE/RESPOND TO THIS QUESTION, AND SOME FOCUS ON THIS ASPECT SHOULD BE REPORTED in the CONCLUSIONS, otherwise the title should be changed.
Abstract:
THIS IS POORLY INFORMATIVE ON THE SPECIFIC GOAL/DISCUSSIONS OF THE present Review in comparison to others already published, and instead should indicate AT THE END what is the original information/novelty and conclusions that the present Authors address in their Review.
Introduction:
-THE REGENERATION OF DIGITS IN CHILDREN OR YOUNG MICE OR OTHER MAMMALS ALSO OVERLAPS WITH THEIR PHYSIOLOGICAL GROWTH AND THEREFORE IS NOT A TRUE REGENERATION BUT A WOUND-HEALING RESPONSE ASSOCIATED WITH A CONTINUOUS GROWTH. I URGE THE PRESENT AUTHORS TO CONSIDER THIS STILL POORLY ADDRESSED ISSUE IN THE EXISTING LITERATURE.
-LINE 41:….Based on these observations, regeneration…..I AGREE WITH THE AUTHORS THAT THE ABILITY TO REGENERATE IS AN ANCIENT PROCESS IN VERTEBRATES, AND THIS HAS BEEN RECENTLY WELL DOCUMENTED BY SOME FOSSIL RECORD AND FROM A DETAILED HYPOTHESIS THAT PROVIDES THE BIOLOGICAL EXPLANATION –THE MOLECULAR DETAILS WILL BE WORKED OUT IN FUTURE STUDIES-FOR THE LACK OF REGENERATION IN TERRESTRIAL VERTEBRATES IN RELATION TO THE LOSS OF THE LARVAL STAGE AND METAMORPHOSIS DURING EVOLUTION (Alibardi, L., 2019. Organ regeneration evolved in fish and amphibians in relation to metamorphosis….. land transition. Annals of Anatomy 222: 114-119.; Alibardi L. 2020. Appendage regeneration in anamniotes utilizes genes active …. J Morphol DOI: 10.1002/jmor.21251JM).
-IN FIG. 1 THE YELLOW LINES in the cladogram FOR CARTILAGINOUS FISH AND CYCLOSTOMES ARE UNCERTAIN (see PM Birds 1978: Tissue regeneration….COPEIA 1978 (2) 345-349; Marconi et al. 2020. Adult chondrogenesis…..eLife 9, e53414, 2020 but other old and poorly known studies on chondricthyes and lamprey may be found through a vast search, SEE MARON K 1960, REGENERATION OF THE TAIL IN LAMPRETRA….. FOLIA BIOLOGICA 8, 55-57, AND ZANANDREA G. 1956: SULLA RIGENERAZIONE DEI PETROMIZONTI…Atti Accadem Sci Istit di Bologna , serie XI (tomo III) 1-36). AS a personal note: IT IS TYPICAL OF JUNIOR RESEARCHERS TO RELY ONLY ON RECENT REFERENCES, GAINED MAINLY BY ELECTRONIC-SEARCH, BUT I ADVICE THE PRESENT AUTHORS TO CONSIDER ALSO OLD STUDIES, UNFORTUNATELY PUBLISHED IN NON-ELECTRONIC AVAILABLE JOURNALS, BEFORE MAKING DEFINITIVE STATEMENTS.
-LINE 82:….immune response to injury: Please consider also:
-Vitulo et al. 2017. Down-regulation of lizard immuno-genes ….an immuno-privileged organ. Protoplasma. 254: 2127-2141.
-Degan et al. 2020. Gene expression in regenerating and scarring …tail regeneration. Protoplasma doi.org/10.1007/s00709-020-01545-6
-Xu et al., 2020. Transcriptome analysis….Anolis carolinensis. J Immunol Reg Medic 7, 100025.
-line 97: FROM which species a genome of 120 Gb has been reported?
-THE SENTENCE:……made lizards one of the first ?? animal models….:PLEASE INDICATE SOME REFERENCE THAT DEMOSTRATE THIS STATEMENT SINCE LIZARDS HAVE ALWAYS BEEN A NEGLECTED MODEL OF REGENERATION, ALSO IN THE PAST.
-Table 1: SINCE ALSO THE GENOME OF THE ITALIAN(EUROPEAN) LIZARD P. MURALIS HAS BEEN PUBLISHED I ADVICE THE AUTHRORS TO INCLUDE IT IN THIS TABLE ( see Andrade et al., Regulatory changes….wall lizard, PNAS 116, 5633-5642, 2019 and their supplementary material, AND also check the NCBI DATABASE)
-line 149 and Fig. 3: it is NOT sure that an AEC is formed in lizards, although one or multiple apical epidermal peg/-s (AEP) are formed, but their morphogenetic potential is not known, see Ref 149.
-FIG. 3:….in response to AEC signals, the central canal…..blastema: TO THE BEST OF MY KNOWLEDGE NO EXPERIMENTAL STUDY HAS INDICATED THIS PROCESS that STILL REMAINS A GOOD GUESS. OTHERWISE, PLEASE, CITE THE SPECIFIC STUDY.
-line 189: …Despite the role of blastema….: OTHER RECENT PAPERS HAVE ANALYZED THE DERIVATION OF BLASTEMA and CARTILAGINOUS CELLS IN LIZARDS, THAT Table 2 does not reflect/consider:
-Alibardi L. 2015. Immunolocalization indicates …e stem/progenitor cells. Micr Res Techn 78: 1032-1045.
-Alibardi L. 2015. Original and regenerating lizard tail cartilage …Micron 78: 10-18.
-Alibardi L 2018. Ultrastructural analysis of ….dedifferentiation … regenerative blastema. J. Morphol. 279: 1171-1184.
-Alibardi L. 2019. Temporal distribution of 5BrdU-labeled cells …..e tail and limb in lizard. Acta Zool 100: 303-319.
ACTIVATION OF THESE CELLS IS ALSO DEMOSTRATED FROM THE ON-EXPRESSION OF TELOMERASE AND C-MYC AFTER 1-3 DAYS POST-AMPUTATION IN MIGRATING CELLS THAT FORM THE BLASTEMA IN LIZARDS.
-line 212:…..Differences are also seen….: THE DETAILED STUDY BY COX (Cox, P.G. 1969. Some aspects of tail regeneration in ……and autoradiography. J. Exp. Zool. 171: 127-150) AND THE OTHERS REPORTED IN THE TEXT, ONLY INDICATE THAT MOST PROLIFERATION OCCURS IN EARLY DIFFERENTIATING TISSUES NOT IN THE BLASTEMA. HOWEVER THIS AND OTHER STUDIES INDICATED THAT ALSO THE LIZARD BLASTEMA CONTAINS PROLIFERATING CELLS (Alibardi L. 2017. Cell proliferation in the ….regenerating tail. Acta Zool 98: 170-180).
THE FOLLOWING STUDIES USING IMMUNOLABELING FOR PCNA (CITED) DETECTED A HIGHER LABELING BUT THIS DEPENDS FROM THE BROADER WINDOW OF PROLIFERATIVE LABELING FOR PCNA IN COMPARISON TO THYMIDINE. SO THEY BOTH INDICATE PROLIFERATION BUT NOT AS INTENSE COMPARING THE AUTORADIOGRAPHICAL studies (MORE RELAIBLE FOR THE S-PHASE THAN USING PCNA) VS the IMMUNOHISTOCHEMICAL DETECTION.
-Line 357:….macrophages in regeneration.: SPECIFIC STUDIES IN LIZARDS HAVE INDICATED THAT M2-LIKE MACROPHAGES ARE PRESENT in the TAIL BUT M1-LIKE IN THE LIMB, see:
- Alibardi L. 2016. Immunolocalization of … and macrophage infiltration in the scarring limb of lizard after limb amputation. Tiss Cell 48: 197-207.
- Alibardi L. 2020. Autoradiography and inmmunolabeling ……arginase-positive M2-like macrophages…. Ann Anat 231 (2020) 151549.
-Line 383-384:….antimicrobial peptides……: THE IMMUNE ACTION ON LIZARD AND FROG REGENERATION HAS BEEN ESTENSIVELY EXAMINED IN:
- Alibardi L 2014. Histochemical, Biochemical ….tissue regeneration. Progr Histoch Cytoch 48: 143-244.
- Alibardi L. 2016. Microscopic observations show invasion …frog….apical epidermal cap and impede regeneration. Ann Anat 210: 94-102.
- Alibardi L. 2017. Review: Hyaluronic acid in the tail and …….immuno-suppressive properties. J Exp Zool B 328: 760-771.
-line 414:…….mid -90s, when Singer….. THIS STATEMENT IS QUITE UNCORRECT (maybe a typo? -90s instead of -50s?) AS MARCUS SINGER SINCE 1952 SHOWED THE IMPORTANCE OF THE NERVE STIMULATION IN AMPHIBIANS AND IN THE FOLLOWING YEARS UNTIL HIS RETIREMENT (AROUND 1988). THE FAMOUS SYNTHESIS OF HIS WORK, THE 1978 REVIEW ON THE “NEUROTROPHIC PHENOMENON….” (citation 123) , INDICATING THE POSSIBLE PROTEINS INVOLVED.
-LINE 429:……then secreted by epithelial cells……: WHAT IS THE TARGETS FOR THE OLOCRINE SECRETION?
-LINE 430:……THE IMPORTANCE OF THE EPENDYMA IN LIZARDS WAS ALSO DEMONSTRATED IN 1987 (Alibardi et al. 1988 , Morphology of experimentally produced tails in lizards. Acta Embr. Morph. Exper. n.s. 9, 181-194) BY IMPLANTS OF THE REGENERATED EPENDYMA AND A TEM ANALYSIS THAT REVEALED PRESENCE OF FEW SPECIFIC NEURONS among ependymal cells (see the recent Alibardi L. 2019. Review: Cerebro Spinal Fluid Contacting Neurons (CSFCNs) …. mechanoreceptors. J Morphol 290: 1292-1308).
-line 432:…..requirement of nerve fibers…..: HOWEVER THE FEW STUDIES CARRIED OUT ON LIZARDS SHOWED THAT THE INNERVATION by periphera nerves WAS NOT A KEY FACTOR FOR the REGENERATION OF LIMBS or TAIL.
-line 477:…..FGF1 and FGF2…..: FGFs ARE ABSENT FROM THE WOUND EPIDERMIS SCARRING LIZARD LIMB BUT PRESENT IN THE TAIL AND WHEN ADMINISTRED TO LIZARD THEY CAN INDUCE THE REGENERATION OF FEMUR, TIBIA AND FIBULA BUT NEVER OF AN AUTOPODIUM:
-Alibardi L. 2012. Observations on FGFs immunoreactivity in the FGFs are required for regeneration. Belg J Zool 142: 23-38.
- Alibardi L. 2017. FGFs treatment on amputated limbs stimulate the regeneration of long …..study. J Morphol Phys Kinesiol 2, 25 doi: 10.3390/jfmk2030025.
- Alibardi L. (2019). Stimulation of regenerative blastema …regeneration in amniotes. Histol Histopath 34: 1111-1120.
-SENTENCES BETWEE LINES 602 AND 611 ARE NOT CLEAR AND SHOULD BE BETTER EXPLAINED.
-LINE 620: IN RELATION TO METAMORPHOSIS TWO PAPERS SHOULD BE CONSIDERED AS THIS IS A KEY ISSUE IN RELATION TO VERTEBRATE REGENERATION:
-Alibardi, L., 2019. Organ regeneration evolved in fish and amphibians in relation to metamorphosis….. land transition. Annals of Anatomy 222: 114-119.
- Alibardi L. 2020. Appendage regeneration in anamniotes utilizes genes active …. J Morphol DOI: 10.1002/jmor.21251JM
-The CONCLUSION SECTION IS NOT WELL ORGANIZED INTO A LOGICAL DISCUSSION. THE TWO KEY PATHWAYS SUSTAINING REGENERATION IN DIFFERENT VERTEBRATES (FGF AND WNT) should BE better emphasized and a more CONVINCING RESPONSE TO THE QUESTION OF THE TITLE (…..WHAT MAKES THIS POSSIBLE?) MORE CLEARLY INDICATED. THIS SHOULD INCLUDE SOME OTHER CONSIDERATIONS ON MAMMALIAN REGENERATION AND ITS FAILURE, AND ON THE MEDICAL ATTEMPTS TO IMPROVE IT, A TOPIC THAT IS LITTLE CONSIDERED IN THE PRESENT REVIEW.
Round 2
Reviewer 2 Report
I have checked the MS and found that the authors have corrected almost all my remarks and the MS contains original informations in addition to those already established in the literature. It will be a nice addition to the existing literature on the topic of regeneration of vertebrates.